# Changes in Psychoacoustic Recognition and Brain Activity by Types of Fire Alarm

**DOI:** 10.3390/ijerph18020541

**Published:** 2021-01-11

**Authors:** Sunghwa You, Woojae Han, Han-Jin Jang, Ghee-Young Noh

**Affiliations:** 1Laboratory of Hearing and Technology, Research Institute of Audiology and Speech Pathology, College of Natural Sciences, Hallym University, Chuncheon 24252, Korea; shyouuu@gmail.com; 2Division of Speech Pathology and Audiology, College of Natural Sciences, Hallym University, Chuncheon 24252, Korea; 3Department of Digital Contents, School of Digital Contents Convergence, Hallym University, Chuncheon 24252, Korea; ff7cloud5@naver.com; 4School of Media Communication, Hallym University, Chuncheon 24252, Korea; gnoh@hallym.ac.kr

**Keywords:** warning signal, arousal, urgency, immersion, electroencephalography

## Abstract

In public, the role of a fire alarm is to induce a person to a certain recognition of potential danger, resulting in that person taking appropriate evacuation action. Unfortunately, the sound of the fire alarm is not internationally standardized yet, except for recommending the use of a signal with a regular temporal pattern (or T-3 pattern). To identify the effective alarm sound, the present study investigated a relationship between acoustic characteristics of the fire alarm and its subjective psychoacoustic recognition and objective electroencephalography (EEG) responses for 50 young and older listeners. As the stimuli, six different types of alarms were applied: bell, slow whoop, T-3 520 Hz, T-3 3100 Hz, and two simulated T-3 sounds (i.e., 520 and 3100 Hz) to which older adults with age-related hearing loss seemed to hear. While listening to the sounds, the EEG was recorded by each individual. The psychoacoustic recognition was also evaluated by using a questionnaire consisting of three subcategories, i.e., arousal, urgency, and immersion. The subjective responses resulted in a statistically significant difference between the types of sound. In particular, the fire alarms had acoustic features of high frequency or gradually increased frequencies such as T-3 3100 Hz, bell, and slow whoop, representing effective sounds to induce high arousal and urgency, although they also showed a limitation in being widely transmitted and vulnerable to background noise environment. Interestingly, there was a meaningful interaction effect between the sounds and age groups for the urgency and immersion, indicating that the bell was quite highly recognized in older adults. In general, EEG data showed that alpha power was decreased and gamma power was increased in all sounds, which means a relationship with negative emotions such as high arousal and urgency. Based on the current findings, we suggest using fire alarm sounds with acoustic features of high frequencies in indoor and/or public places.

## 1. Introduction

### 1.1. Background of Fire Alarms

With increasing demands of human ergonomics for auditory warning signals and advancing technologies in the engineering for their application, signals have been developed. Remarkably, the human ergonomic guidelines for the design of auditory warning signals proposed by Patterson [1] stimulated various and extensive studies. Researchers had wanted to understand a relationship between acoustic features of the warning signals and subjective responses of the listeners, while focusing on components such as frequency, intensity, and rate [2,3,4]. Among the warning signals, the fire alarm has also been studied for several decades [5], but the alarm is not unified or standardized yet beyond the rough suggestion of a temporal pattern.

The temporal three (T-3) pattern is recommended by the International Organization for Standardization (ISO) as an international fire alarm sound [6]. Additionally, this pattern is suggested by the United States National Fire Protection Association 72 (1996) [7] and The National Building Code of Canada (1995) [8]. To be specific, the T-3 pattern consists of signaling on/off for three repetitions of 0.5 s, and then the signal is off after being maintained for 1.5 s (please see Figure 1). Such a regular T-3 pattern is noticeable in background noise, and thus can minimize the masking effect from the fire alarm [9]. However, this pattern has a flaw in being commonly applied to any kind of sound because of the existence of only a time-domain without acoustically specific features. Furthermore, the frequency range of the T-3 pattern is different in manufacturers [10,11], which warrants further study to determine a scientific and evidence-based fire alarm [11]. On the other hand, a bell has often been used as a fire alarm for public buildings in Korea and Canada, and a slow whoop is mainly used in buildings in the U.S. [9]. The slow whoop proposed by Gosswiller [12] is defined as a square wave tone with a frequency range of 500 to 700 Hz. It is characterized by being gradually increased by one octave for 2.5 to 4 s at the base frequency and then is repeated after a brief pause of 0.3 to 0.5 s. Slightly differently, the slow whoop of Humphreys [13] gradually increases in frequency from 600 to 1100 Hz, with a short cycle of about 4 s. However, it also emphasizes intensity and/or signal-to-noise ratio while being used in a few standards of the U.S., U.K., and South Korea without considering the frequency-specified information.

Since the human ear is more sensitive in the mid-frequency than either lower or higher frequencies, it simply seems effective for fire alarms to have frequencies between 1000 and 3000 Hz [14]. On closer scrutiny, in terms of acoustics, fire alarms with high-frequency limits can be transmitted widely, whereas the fire alarm with low frequency, i.e., below 1000 Hz, has advantages by diffraction when meeting obstacles or walls [14]. Moreover, the high-frequency fire alarm is not appropriate for listeners with hearing loss, especially poor hearing ability in the high frequencies due to aging. For example, Bruck and Thomas [15] presented several frequencies of a T-3 fire alarm (500 to 3000 Hz) to sleeping older adults. Their result showed that high-frequency fire alarms needed 20 dBA higher levels than those of low frequency to wake the sleeping older adults. It is therefore persuasive to use the fire alarm with the acoustic feature of low-to-mid-frequencies. Nevertheless, results can be quite contrary when we consider listeners’ subjective cognition. In the study conducted by Proulx et al. [9], the low-frequency T-3 was perceived as either mechanical or a ringtone rather than fire alarms, suggesting that the bell and slow whoop were better as a fire alarm with sufficient urgency. Additionally, Kahn [16] and Tong and Canter [17] reported that the T-3 was not recognized as an emergency signal because the listeners were already familiarized with similar alarms, which may not be successful to induce evacuations. Lee [18] agreed that T-3 composed of low frequencies showed lower urgency than T-3 of high frequency.

### 1.2. Fire Alarms and Psychoacoustic Recognition

As its function, the fire alarm should alert people to potential danger accurately and quickly, consequently resulting in the recognition of the danger instinctively, allowing fast and safe evacuation to occur [19]. To evaluate psychoacoustic aspects that people could perceive during the fire alarm, arousal, urgency, and immersion are considered. First, arousal is a psychological state in which an external stimulus causes surprise or interest and curiosity [20]. Because it indicates an emotional change in the autonomic nervous system from the external stimuli, the arousal level varies depending on the type of stimulus [21]. When arousal is applied to the sound of a fire alarm, Bruck and Thomas found that higher arousal can cause greater cognitive evocation, which may reduce reaction time for sleeping residents during a fire [10]. Second, a sense of urgency to a perceived threat or fear causes a response to the vulnerability. Thus, it might refer to the degree of perception of a dangerous situation [22], explaining that the greater urgency of external stimuli, the greater the risk response rate and accuracy [23]. When applied to the fire alarm, we suppose that the urgency differs in acoustical characteristics of the fire alarm, and communicates accurately whether there is a potential danger or risk. Third, immersion is defined as mental and physical excitement due to a mind that is absorbed by external stimuli [24]. In the previous studies, strong immersion into the external stimuli enhanced emotions such as cognition [25]. It was also confirmed that the degree of immersion revealed a static correlation with anxiety and fear when the stimulus that induced an affective process such as anxiety, fear, and depression was presented [26].

### 1.3. Fire Alarms and Human Brain Activity

During the past 20 years, many electroencephalography (EEG) studies have provided insights into the neural variations in emotions [27,28,29]. Thanks to their high temporal resolution, neural processes at different time scales are possible to assess. Traditionally, frequency bands are related to different mental functions and presumably, also to most emotional states as well. The EEG power spectrum is divided into five frequency bands: delta, theta, alpha, beta, and gamma [30].

In detail, theta corresponds to the frequency band of 4–8 Hz, with two manifestations [31]. The first type of theta is observed in widespread distribution and thus is related to drowsiness and stated of low-level arousal such as inefficient information processing. The second type that is observed in the frontal midline of the cerebral is considered related to cognitive demands such as calculation [31,32], and learning and error processing [33,34]. Thus, it could be interpreted as a high mental effort and attention. Additionally, the expression of theta in the central frontal lobe has been reported to be derived from the anterior cingulated cortex (AAC), which corresponds to emotion [35], while providing increased theta for pleasant sounds [36] and decreased theta power for unpleasant sounds [37]. On the other hand, alpha, first reported by Berger [38], has a frequency band of 8–13 Hz. It is usually observed in the parietal and occipital region, called posterior rhythm [39]. In addition, it appears in the preparation of movement in the somatosensory cortex [40] and appeared in the auditory cortex of the temporal lobe [41]. The alpha power is well known as EEG that appears primarily with a closed eye. Because the alpha power is increased under comfortable conditions such as rest and relaxation [41] and decreased during negative emotions such as anxiety and danger [42,43], it shows a negative correlation with brain activity.

Generally, beta is related to arousal and observed in the frontal or temporal lobe [40]. Widely distributed beta (13–30 Hz) of EEG is thought to be related to increased alertness and cognitive processes [40]. In the present study, beta was subdivided by SMR (sensorimotor rhythm; 12–16 Hz), mid-beta (16–21 Hz) and high-beta (21–30 Hz). Several studies that investigated the relationship between beta and emotions confirmed that increased arousal corresponded to increased beta [36]. That is, the SMR is associated with the somatosensory cortex of the previous alpha, and thus related to the attention to the external stimuli [39]. Similarly, mid-beta is also related to brain activities such as attention and arousal. Finally, the power of high-beta is increased with brain activities such as attention, arousal, and learning. On the other hand, gamma corresponds to the frequency band of 30–50 Hz, while being observed in the frontal lobe. While explaining some brain pathological problems, extreme arousal, anxiety, and excitement, the increased arousal or anxiety could correspond to the increased gamma power [44].

Again, fire alarms should alert listeners to potential hazards and prompt them to make the right decision, such as evacuation. Unfortunately, current suggestions or regulations regarding fire alarms do not include sufficient and scientific evidence. It is only found that low-frequency fire alarms have benefits in terms of acoustics while high-frequency fire alarms have benefits in terms of subjective perception. Therefore, the present study aimed to evaluate both subjective (psychoacoustic recognition) and objective (electroencephalography; EEG) responses using six types of fire alarms with various acoustic characteristic and also to suggest the most effective alarm sound to be used for the general public.

## 2. Materials and Methods

### 2.1. Participants

While G-Power program was used to calculate an appropriate sample size for the groups [45], a total of 50 subjects who are local residents, 25 young adults (M: 25.1, SD: 1.58, range: 23–28) and 25 older adults (M: 74.48, SD: 3.70, range: 68–83), participated in the study voluntarily. Each group consisted of 13 males and 12 females. Inclusion criteria were as follows: not have any otological and/or psychiatric history, not have chronic otologic and neurological diseases without taking any meditations, and have normal hearing thresholds in the testing frequencies of 250 to 8000 Hz as a function of age based on ISO-7029 [46] (See Table 1 for detailed information). Further, all participants conducted the Korean version of the Mini-Mental State Examination (MMSE-K) before beginning the experiment and they confirmed to have a normal cognitive function with 25 scores or greater of MMSE-K. All participants were native Koreans and signed an informed consent form. The procedures were approved by the Institutional Review Board of Hallym University (#HIRB-2020-01).

### 2.2. Development of Six Types of Fire Alarm Stimuli

Based on the National Fire Protection Association 72 (1995) and The National Building Code of Canada (1995) [8], three alarm sounds that are typically used as a fire alarm, bell, slow whoop, and T-3 pattern, were chosen. Bruck and Thomas [10] reported that a general smoke detector that adopted a T-3 pattern had dissimilar acoustic characteristics across the manufacturers. Although T-3 3100 Hz had been frequently used, T-3 520 Hz squared wave was effective for sleeping residents, older adults, and even for patients with hearing loss [10]. Thus, we selected two different T-3 patterns, namely “T-3 520 Hz” and “T-3 3100 Hz” while considering specific acoustic characteristics of T-3 pattern (please see Table 2 and Figure 2A–D for their detailed acoustic features).

When a listener has hearing loss due to aging, he/she has certain disadvantages to perceiving the fire alarm in terms of objective and subjective recognition. In the ISO-7029 [46], the older adults have higher thresholds in high frequencies than their young counterparts even though they have normal hearing for their age. This means that the fire alarm sounds with acoustic features of high frequencies (e.g., bell, T-3 3100 Hz) provide reduced sound intensity to them, consequently resulting in a difficulty in correctly recognizing the sound and subsequently evacuating. In view of that, we manipulated frequency-dependent attenuation to the fire alarms of both T-3 520 Hz and T-3 3100 Hz—namely, “simulated T-3 520 Hz” and “simulated T-3 3100 Hz,” respectively, while using a graphic equalizer function of Adobe^®^ Audition CS6 (Adobe^®^, San Jose, CA, USA) based on the standardized hearing thresholds updated for the aging (ISO-7029) [46]. In detail, modulation and attenuation were performed in 30 frequency bands between 250 and 12,500 Hz. As a result, there was the attenuation for the harmonics at about 1500 Hz and above in the simulated T-3 520 Hz (Figure 2E), whereas the simulated T-3 3100 Hz had amplitude reduction and modulation in the high-frequency bands above 3100 Hz (Figure 2F).

Except for the bell, five fire alarm sounds were generated based on the definition of each sound source through the MATLAB R2019b software (MathWorks, Natick, MA, USA). The bell was restricted to being generated through typical mathematical formulas because of sounds generated by repeated physical impact. Thus, we recorded a sound of the 24 V warning bell that was generally provided in domestic fire hydrants in a sound ambient room through Computerized Speech Lab (CSL; Model 4150B, KayPentax, Montvale, NJ, USA). The acoustic characteristics of all fire alarm sounds were summarized in Table 1 and Figure 2.

### 2.3. Development of the Subjective Questionnaire

While considering the purpose of a fire alarm, a questionnaire was developed to evaluate three psychoacoustic aspects that people could perceive when listening to the alarm sound: arousal, urgency, and immersion.

For the arousal subcategory, arousal items from Hoffman and Novak [24] were adopted via translation and back-translation from English to Korean and reconstructed 4 questions related to the fire alarms [24]. Assessing urgency to the fire alarm sounds was intended to confirm by two factors with 4 items. That is, it included evaluating a degree to which rapid situational awareness (emergency) and resolution (critical) was required, which could pose a significant risk [22]. Finally, because the fire alarm sounds with immersion could induce two factors, e.g., variability and enhancement of the risk perception, 2 items per each factor were adapted from Beck et al. [26]. A total of 12 items (3 subcategories × 4 items) were subjectively assessed via a 7-point scale, which was the most appropriate scale for the subjective evaluation [47]. The developed questionnaire is provided in Appendix A.

### 2.4. EEG Data Acquisition

Based on the international 10–20 system [48], EEG was recorded onto 21 sites using DSI-24 device (Dry Electrode EEG Headset; Wearable Sensing, San Diego, CA, USA). Among 21 integrated electrodes for the EEG data acquisition, only 16 electrodes (i.e., Fp1, Fp2, Fz, F3, F4, F7, F8, T3, T4, T5, T6, Pz, P3, P4, O1, O2) were used for the analysis while ignoring 3 unrelated sites (Cz, C1, C2) and references (A1, A2) for artifacts control. The EEG data were acquired at a sampling rate of 512 Hz, with electrode impedances below 10 KΩ for artifact rejection. During the experiment, the subjects were asked to rest in a relaxing chair, and the EEG signals were acquired for 60 s while listening to the fire alarm. Each subject conducted 7 sections by random stimuli order (1 for no sound condition as the background and 6 times for each fire alarm).

### 2.5. Experimental Procedures

Both subjective questionnaire evaluation and objective EEG measurement were performed for 6 different fire alarm sounds per each participant. All experiments were conducted in a sound ambient room. Each fire alarm was presented for 35 s from a loudspeaker (Sound Field Speaker, Eden prairie, MN, USA) located at 0° at 1 m distance from each subject. Since different types of the fire alarm might create different levels of sound pressure, we made sure to present the same levels across the stimuli while equally calibrating the presentation levels. To calibrate the stimuli, a sound level meter (Type #2250, Bruel and Kjær, Nærum, Denmark), coupled with a ½-inch free field microphone (Type #4189, Bruel and Kjær, Nærum, Denmark), was used to measure the intensity of the stimulus level from 20 Hz to 20 kHz, which is the audible frequency range of the human ear. The intensity level of the fire alarm was 75 dBA.

To minimize crossing errors, presentation order of the fire alarms was pseudo-randomized. Before listening to the fire alarm sounds, background EEG was measured for all subjects for 35 s in a relaxed condition without any external stimuli as the individual EEG reference. Then, six sounds of the fire alarm were presented by connecting the GSI-61 (Grason-Stalder, Eden prairie, MN, USA) to the laptop (6th Generation Thinkpad X1 Carbon, Lenovo Group Ltd., Beijing, China) (Figure 3) and EEG responses of each stimulus per subject were recorded. Notice that the subject was constructed to limit any physical movement while listening to the signals in the relaxing chair with a neck pillow.

After conducting the EEG data on the fire alarm sound, a questionnaire of 12 items was distributed to evaluate the amount of arousal, urgency, and immersion. The same procedure was repeated for six fire alarms, with a 3-min break between test intervals to minimize the effect of the previously presented fire alarm [44]. In addition, to minimize the cross error of the questionnaire due to repeated measures, the items of the questionnaire were organized randomly in each evaluation.

### 2.6. Data Analysis and Statistical Analysis

After EEG acquisition, all EEG signals were filtered into 0.2–50 Hz band-pass and saved comma-separated values (CSV) files. The CSV files were pre-processed and converted to a CDF extension using Batch-PRO software. The converted CDF files were analyzed for post-processing using the Telescan (Laxtha Inc., Korea). During post-processing, the raw data analyzed EEG variations by frequency range through Fast Fourier Transformation (FFT) analysis. Additionally, the power spectrum of each frequency was used to calculate the theta (4–8 Hz), alpha (8–13 Hz), SMR (12–16 Hz), mid-beta (16–21 Hz), high-beta (21–30 Hz) and gamma (30–50 Hz). Then, the power spectrum extracted EEG data by the Absolute Band Power Parameter method. Data by waveform at each frequency range extracted through all previous processes were classified into background EEG (quiet) and EEGs by six fire alarms. To calculate the EEG fluctuation due to listening to fire alarms, background EEG was subtracted from EEG measured by a fire alarm (background EEG—fire alarm EEG). Furthermore, the upper and lower 5 s of the 35 s were removed for upper- and lower-limit, respectively, resulting in getting rid of upper and lower 5 percentiles from the data. Finally, the analysis included only EEG dates for a total of 5–30 seconds’ 90 percentile dates.

All statistical analysis in the present study was performed with SPSS software (Version 20; IBM Corporation, Armonk, NY, USA) with *p* < 0.05 criteria. To identify the reliability and validity of the developed questionnaire, Cronbach’s α and principal components analysis (PCA) with varimax rotation were conducted. Measured subjective recognitions (arousal, urgency and immersion) and EEG (theta, alpha, mid-beta, high-beta and gamma) were analyzed by a three-way analysis of variance (ANOVA). Specifically, the three-way ANOVA was performed to confirm the interaction among the main effects of type of fire alarms (frequency), age, and gender. As a post-hoc analysis, Bonferroni multiple comparisons were applied.

## 3. Results

### 3.1. Development of Questionnaire for Psychoacoustic Recognition

#### 3.1.1. Reliability of the Questionnaire Items

The psychoacoustic and subjective recognition of respondents was analyzed in the questionnaire on three factors that can be induced for listening to fire alarms: arousal, urgency, and immersion. Specifically, the questionnaire evaluated three factors by 12 items: four items per subcategory. As a result, reliability showed high reliability and indicated that they are homogeneous items: arousal (α: 0.872), urgency (α: 0.898) and immersion (α: 0.953).

#### 3.1.2. Principal Component Analysis of the Questionnaire

The PCA was conducted to verify that three factors used to measure psychoacoustic recognitions using subjective responses were statistically assigned and accurately measured. According to the analysis, the Kaiser–Mayer–Olkin (KMO) test was 0.801 (*p* < 0.000), which explained with significance that the questionnaire was quite appropriate for exploring the factor structure. Additionally, Bartlett’s test of sphericity confirmed that the correlation matrix was significantly different from the identity matrix (χ^2^: 401.878, *p* < 0.000). When the factor was extracted through PCA, the components extracted principal axis factoring and varimax rotation with more than 1.0 eigenvalues. Then, only more than 0.05 loaded factors were included. As a result, three factors were extracted in the developed questionnaire. Each eigenvalue was 3.738 (factor 1), 2.024 (factor 2), and 1.882 (factor 3) and a total of the variances was 72.82.

As described earlier, the questionnaire consisted of 12 questions, divided into three subcategories with four items each: arousal, urgency, and immersion. Appendix A shows that the rotated component matrix was higher than 0.4. As a standard value of the normal range of factor loading, this means that the items interact with each other, and the validity of the items constituted by each factor. Factor 1 assigned five questions, with items 5–8 related to urgency and item 2 related to arousal. Factor 2 confirmed four questions, with items 9–12 related only to immersion. Finally, factor 3 assigned three questions except item 2, with items 1, 3 and 4 related to arousals.

### 3.2. Evaluation of Psychoacoustic Recognition

As a result of the arousal changes caused by listening to fire alarms (Figure 4A,B), the young adult males were evaluated at the highest arousal at the T-3 3100 Hz (mean: 18.15 out of 21 points). Bell (18.08 points), simulated T-3 3100 Hz (18.00 points) and slow whoop (17.54 points) followed high arousal. Although the difference in arousal level was not noticeable among the four fire alarms, T-3 520 Hz and simulated T-3 520 Hz showed relatively lower arousal, 15.77 and 13.77 points, respectively. For adult females, T-3 3100 Hz was reported as the highest arousal at 20.50 out of 21 points, followed by slow whoop (20.00 points), and T-3 520 Hz (17.92 points). Simulated T-3 3100 Hz, bell, and simulated T-3 520 Hz were evaluated with relatively low arousal at 17.42, 16.58, and 14.42 points, respectively. Commonly, the T-3 3100 Hz showed the highest arousal, and simulated T-3 520 Hz evaluated the lowest arousal. In terms of frequency of the fire alarm, the T-3 3100 Hz had relatively high, but the simulated T-3 520 Hz had a lower fundamental frequency. Additionally, as for the comparison with T-3 and simulated T-3, the simulated T-3 520 and 3100 Hz evaluated relatively lower arousal than T-3 520 and 3100 Hz in adults. 

In the older adults, males evaluated the highest arousal for simulated T-3 3100 Hz at 18.69 of 21 points. Additionally, bell, T-3 3100 Hz and slow whoop followed with 17.69, 17.69 and 17.62 points, respectively. Similar to the young adult males, the T-3 520 Hz and simulated T-3 520 Hz, which had low fundamental frequency, evaluated relatively low arousal; 16.85 and 15.46 points. For older adult females, T-3 3100 Hz reported the highest arousal at 18.83 points, followed by bell, slow whoop and simulated T-3 3100 Hz; 18.42, 18.42 and 18.25 points, respectively. The older females also reported relatively low arousal for simulated T-3 520 Hz and T-3 520 Hz, 16.08 and 15.50 points, respectively.

As a result of three-way analysis variance, arousal had significant main effects depending on the type of fire alarm (F: 6.845, *p* < 0.01), but the main effects of age (F: 0.063, *p* = 0.801) and gender (F: 1.696, *p* = 0.194) were not significant (Table 3). The partial ŋ^2^ of the type of fire alarm was 0.110, which indicates a moderate effect size. Additionally, the interactions among fire alarm, age- and gender-related variables were not significant. With post-hoc analysis of the type of fire alarm, T-3 3100 Hz, which showed the highest arousal, had significantly higher arousal than T-3 520 Hz and simulated T-3 520 Hz. Additionally, slow whoop, simulated T-3 3100 Hz and bell reported significantly higher arousal than simulated T-3 520 Hz. When simulating to the young adults for presbycusis effect, the results of a paired t-test of T-3 (520 and 3100 Hz) and simulated T-3 (520 and 3100 Hz) did not confirm a significant difference between 520 Hz (t: 1.457, *p*: 0.171) and 3100 Hz (t: 0.187, *p*: 0.855) in males. However, the females showed a significant difference between 520 Hz (t: 3.251, *p* < 0.05) and 3100 Hz (t: 2.475, *p* < 0.05).

In the case of urgency according to the type of fire alarm (Figure 4C,D), adult males had the highest urgency at slow whoop (mean: 18.15 of 21 points), followed by bell, simulated T-3 3100 Hz, and T-3 3100 Hz as 17.08, 16.92, and 16.38 points, respectively. The T-3 520 Hz and simulated T-3 520 Hz which showed low fundamental frequency were evaluated as relatively low urgency as 15.23 and 10.54 points, respectively. An adult female showed the highest urgency with T-3 3100 Hz at 19.92 points, followed by slow whoop, T-3 520 Hz and simulated T-3 3100 Hz at 19.50, 15.67 and 15.43 points, respectively. In addition, bell and simulated T-3 520 Hz reported relatively low urgency at 13.58 and 12.08 points, respectively. Considering the adult male, the T-3 520 Hz, which had low fundamental frequency, evaluated relatively higher for females than males. In terms of older adults, the older adult male evaluated as the highest urgency (21.67 points) at simulated T-3 3100 Hz, followed by slow whoop, T-3 3100 Hz, and bell at 21.85, 21.62, and 20.85 points, respectively. Similar to the adult male, the older adult male showed relatively low urgency at T-3 520 Hz and simulated T-3 520 Hz (low fundamental frequency); 17.62 and 17.16 points, respectively. The older female evaluated the bell as the highest urgency at 21.67 points, followed by simulated T-3 3100 Hz, T-3 3100 Hz and slow whoop at 19.92, 19.83 and 19.33 points, respectively. Additionally, they reported T-3 520 Hz and simulated T-3 520 Hz (low fundamental frequency) as relatively low urgency at 16.83 and 14.58 points. 

As a result of the three-way analysis variance, the main effects of type of fire alarm (F: 10.119, *p* < 0.000) and age (F: 36.843, *p* < 0.000) were significant, but the main effect of gender was not confirmed (F: 1.038, *p*: 0.039; Table 4). The partial ŋ^2^ by type of fire alarm and age was identified at 0.115 and 0.118, respectively. This means a moderate or large effect size. Similar to arousal, the interaction of all variables was not significant. Specifically, the slow whoop and T-3 3100 Hz confirmed significantly higher than T-3 520 Hz and simulated T-3 520 Hz. Additionally, bell and simulated T-3 3100 Hz confirmed to significantly higher urgency than simulated T-3 520 Hz. As a result of post-hoc analysis for age, the older adults showed significantly higher urgency levels than the young adult group. In terms of presbycusis simulation, a paired t-test for T-3 (520 and 3100 Hz) and simulated T-3 (520 and 3100 Hz) confirmed significant differences (t: 3.184, *p* < 0.05) in 520 Hz among males. However, 3100 Hz confirmed no significant differences (t: −0.442, *p*: 0.666). For the female, significant differences were observed at both 520 Hz (t: 2.305, *p* < 0.05) and 3100 Hz (t: 3.822, *p* < 0.05).

As a result of immersion evaluation by listening to fire alarms, adult males gave the highest immersion (mean: 18.92 out of 21 points) for slow whoop (Figure 4E,F). Additionally, T-3 520 Hz, T-3 3100 Hz and bell followed in order of high immersion; 16.62, 16.38, and 16.15 points, respectively. Simulated T-3 3100 Hz and 520 Hz induced relatively low immersion; 16.08 and 15.38 points. Adult females evaluated slow whoop (19.92 points) as the highest immersion, followed T-3 3100 Hz, simulated T-3 3100 Hz and T-3 520 Hz; 17.83, 17.50, and 15.75 points, respectively. Additionally, simulated T-3 520 Hz and bell were evaluated as a low immersion at 15.25 and 13.42 points, respectively. In terms of older adults, older males reported simulated T-3 3100 Hz (23.31 points) as the highest immersion, followed by T-3 3100 Hz, bell, and slow whoop; 23.23, 23.00, 22.62 points, respectively. Relatively, the T-3 520 Hz and simulated T-3 520 Hz were evaluated as low immersion; 21.92 and 21.77. In the older female, bell was evaluated as the highest immersion at 22.83 points, followed by simulated T-3 3100 Hz, T-3 520 Hz, and T-3 3100 Hz at 22.75, 22.33 and 21.75 points, respectively. The slow whoop and simulated T-3 520 Hz reported relatively low immersion at 21.50 and 20.67 points. Overall, immersion cannot confirm the tendency for the low fundamental frequency to correspond to low arousal and urgency.

As a result of the three-way analysis variance, immersion confirmed significant main effects of type of fire alarm (F: 2.867, *p* < 0.05) and age (F: 182.481, *p* < 0.001) as well as urgency (Table 5). The partial ŋ^2^ of type of fire alarm and age was 0.049 and 0.398, respectively. The type of fire alarm showed a small to moderate effect size, and age was identified as a very large effect size. Additionally, unlike other factors, immersion confirmed a significant interaction (F: 2.948, *p* < 0.05) between the type of fire alarm and age with a small-to-moderate effect size (partial ŋ^2^: 0.051). As a result of post-hoc analysis of the type of fire alarm, slow whoop, which showed the highest immersion, confirmed a significant difference with simulated T-3 520 Hz. The results of the paired t-test for immersion of T-3 and simulated T-3 showed no significant difference at both 520 Hz (t: 0.810, *p*: 0.424) and 3100 Hz (t: 0.298, *p*: 0.771). Similarly, no significant difference was found in both 520 Hz (t: 0.589, *p*: 0.568) and 3100 Hz (t: 0.251, *p*: 0.806).

### 3.3. Brain Activity by EEG Measurement

The present study applied EEG to objectively measure emotional changes expressed in response to listening to fire alarms. Based on the frequency band, EEG variables were divided into theta (4–8 Hz), alpha (8–13 Hz), SMR (12–16 Hz), beta (mid-beta; 16–21 Hz and high-beta; 21–30 Hz), and gamma (30–50 Hz) (please see Figure 5).

#### 3.3.1. EEG Data: Theta (4–8 Hz)

In the electrode placement, the theta power (4–8 Hz) over frontal electrodes (Fp1, Fp2, Fz, F3, F4, F7, F8) was greater during listening to fire alarms than other electrodes. As a result, changes of theta power on the listening condition for adult males were highly decreased at T-3 520 Hz (−11.61), bell, T-3 3100 Hz, simulated T-3 520 Hz, and slow whoop also decreased with −5.78, −3.75, −3.38 and −1.69, respectively. On the other hand, simulated T-3 3100 Hz increased in listening conditions than background at 1.16. In terms of adult females, the theta of adult females increased in listening conditions for all fire alarms. Specifically, slow whoop increased the most at 12.62, followed T-3 520 Hz, and simulated T-3 520 Hz at 11.09 and 7.78. The T-3 3100 Hz, simulated T-3 3100 Hz and bell, which had high fundamental frequency, measured relatively small increases at 3.90, 1.19 and 0.15, respectively. In older adults, older males showed increased theta power similar to adult females. Specifically, the T-3 520 Hz increased the most at 18.65, followed by simulated T-3 3100 Hz and T-3 3100 Hz, simulated T-3 520 Hz, and T-3 520 Hz at 14.87, 13.87, 11.94 and 10.81, respectively. The slow whoop measured the least increase at 9.02. In older females, the change of theta power differentiated according to the fire alarm. Simulated T-3 520 Hz, T-3 3100 Hz and slow whoop increased at 17.44, 6.97 and 1.20, respectively. On the other hand, the bell, simulated T-3 3100 Hz and T-3 520 Hz decreased at −18.47, −10.91 and −3.60, respectively. The statistical analysis of theta according to type of fire alarm, age and gender showed that all variables had no significant effect. In other words, the three variables did not have a statistically significant effect on the change in theta. In terms of interaction, interactions between age and gender were significant (F: 8.960, *p* < 0.05) and showed a small effect size (partial ŋ^2^: 0.030). Additionally, other significant interactions between variables were not founded.

#### 3.3.2. EEG Data: Alpha (8–13 Hz)

As a result of confirming the distribution of the alpha power according to the placement of the electrode, the younger group was weak, but the changes of the alpha power increased as the electrode progressed to the occipital area of the brain. However, the overall tendency showed widely distributed responses in all electrodes. The older group also found that the alpha power variations were widely distributed regardless of the placement of the electrodes. The changes in alpha power in listening conditions tended to decrease regardless of age and gender. Specifically, for young males, the alpha decreased the most at −18.56 at simulated T-3 3100 Hz, followed by −16.56 and −15.70 at bell and T-3 520 Hz, respectively. T-3 3100 Hz and slow whoop reduced alpha power at −14.91 and −12.99, and simulated T-3 520 Hz showed the least reduction (−11.02). Young females, similar to males, measured the most reduction in alpha at simulated T-3 3100 Hz (−8.26), followed by T-3 3100 Hz and bell at −5.71 and −4.61. Subsequently, the simulated T-3 520 Hz and slow whoop also decreased to −3.72 and −3.09 respectively; the lowest reduction in alpha was T-3 520 Hz at −2.44. For older males, the slow whoop decreased the most in alpha at −3.04, followed by simulated T-3 520 Hz and T-3 520 Hz at −2.90 and −2.74, respectively. Additionally, the bell and simulated T-3 3100 Hz decreased at −2.43 and −2.08, and the T-3 3100 decreased the least at −1.80. Older females showed the largest decrease in alpha at −6.74, followed by simulated T-3 3100 Hz and simulated T-3 520 Hz at −5.08 and −2.71. T-3 520 Hz and slow whoop, also decreased, at −2.67 and −2.65. The lowest reduction in alpha was identified at T-3 3100 Hz (−1.79), the same as in older males. Statistical analysis of changes in alpha according to listening condition showed that the main effect of type of fire alarm was not significant (F: 0.516, *p*: 0.764). In other words, the acoustic characteristics of the fire alarms did not significantly affect the changes in alpha power. However, the main effects of age (F: 20.014, *p* < 0.000) and gender (F: 9.329, *p* < 0.01) were significant. These results suggest that age and gender were affected by alpha wave changes, with small-to-moderate effect size (partial ŋ^2^: 0.064 and 0.031, respectively). In addition, significant interaction between age and gender was identified (F: 14.358, *p* < 0.000), with small-to-moderate effect size (partial ŋ^2^: 0.047). The post-hoc analysis confirmed a significant difference (young adult: −9.80, older adult: −3.30). It also showed a significant difference for gender (male: −8.96, female: −4.16).

#### 3.3.3. EEG Data: Beta (12–30 Hz); SMR (12–16 Hz), Mid-Beta (16–21 Hz) and High Beta (21–30 Hz)

In the present study, the beta power (12–30 Hz) was subdivided as a frequency domain, such as SMR (sensorimotor rhythm, 12–16 Hz), mid-beta (16–21 Hz) and high-beta (21–30 Hz). As a result of confirming the distribution of the beta power according to the electrode placement, the sub-beta powers were distributed widely regardless of electrode placement. 

For the SMR power, the young male decreased in all listening conditions regardless of the type of fire alarm. Specifically, bell measured the most decrease to SMR power at −1.79, and simulated T-3 3100 Hz, and T-3, 3100 Hz followed at −1.46 and −1.45, respectively. Slow whoop and simulated T-3 520 Hz also decreased at −1.23 and −1.11, respectively, and the T-3 520 Hz identified as a least decreased fire alarm at −1.01. For young females, all fire alarms except T-3 520 Hz identified a decrease. Specifically, slow whoop showed the greatest decrease (−1.01). Simulated T-3 3100 Hz, bell, T-3 3100 Hz, and simulated T-3 (520 Hz) were also decreased; −0.99, −0.79, −0.75, and −0.11, respectively. On the other hand, the T-3 520 Hz increased to 0.27 based on the listening condition. In older males, it was observed an increasing tendency, except slow whoop, unlike young adults, who showed a decreasing tendency to SMR. Among them, the SMR increased the most at 1.12 in bell, followed by T-3 3100 Hz, simulated T-3 520 Hz, T-3 520 Hz and simulated T-3 3100 Hz at 0.78, 0.54, 0.30 and 0.27, respectively. In contrast, the slow whoop was measured at −0.16, which was the SMR power decreased. The older females, similar to young adults, confirmed the tendency to decreased SMR power. Specifically, bell and simulated T-3 3100 Hz showed a relatively high decrease with −1.61 and −1.45, followed by slow whoop and simulated T-3 520 Hz; −1.36 and −1.35. In contrast to the previous fire alarms, the T-3 520 Hz increased the SMR power at 0.27. As a result of statistical analysis for the changes of the SMR power according to the listening condition, the main effect for age was significant (F: 32.882, *p* < 0.05) with small effect size (partial ŋ^2^: 0.02). However, the other main effects according to the type of fire alarm (F: 32,882, *p*: 0.521) and gender (F: 2.218, *p*: 0.137) were not significant. Nonetheless, significant interactions were confirmed for age and gender (F: 111.881, *p* < 0.000) with moderate effect size (partial ŋ^2^: 0.65). As a result of post-hoc analysis according to age, a significant difference (young adult: −0.96, older adult: −0.28) was confirmed within age.

In the mid-beta, the young males showed a tendency to decreased mid-beta power. Specifically, simulated T-3 3100 Hz was the largest decrease at −1.12, followed by T-3 3100 Hz, bell and simulated T-3 520 Hz and T-3 520 Hz at −0.82, −0.81, −0.58 and −0.48, respectively. In particular, slow whoop showed the least decrease. For young females, the T-3 3100 Hz and simulated T-3 3100 H decreased the mid-beta power at −0.19 and −0.08, respectively. However, bell, T-3 520 Hz, slow whoop and simulated T-3 520 Hz tended to increase the mid-beta power at 0.65, 0.53, 0.14 and 0.12, respectively. For older males, slow whoop only decreased at −0.17 in the mid-beta power, and other fire alarms were increased. T-3 3100 Hz showed the largest increase was 0.81, and bell, simulated T-3 520 Hz and T-3 520 Hz followed at 0.51, 0.47 and 0.30, respectively. In addition, simulated T-3 3100 Hz showed the smallest increase at 0.05. Older females, similar to young males, decreased in all fire alarms. Specifically, simulated T-3 3100 Hz, and bell decreased to −1.02 and −0.99 respectively. Additionally, slow whoop, simulated T-3 520 Hz, T-3 3100 Hz and T-3 520 Hz were decreased to −0.45, −0.44, −0.28 and −0.22, respectively. Statistical analysis of changes in mid-beta according to type of fire alarms, age and gender showed that all main effects of type of fire alarm, age and gender were not significant. It means that mid-beta variations had not been affected by the type of fire alarm, age and gender. In terms of interaction, significant interactions between age and gender were identified (F: 19.634, *p* < 0.000), with a moderate effect size (partial ŋ^2^: 0.063). Additionally, all post-hoc analyses were not significant.

As a result of measuring high-beta power for young males, high-beta showed a tendency to reduce in all fire alarms. The slow whoop decreased the most at −0.60, followed by simulated T-3 3100 Hz, simulated T-3 520 Hz and bell at −0.46 −0.34 and −0.33, respectively. For young females, it was found, contrary to males, that all fire alarms tended to increase in high-beta. Specifically, T-3 520 Hz had the largest increase at 1.45, while bell had the smallest increase. Other fire alarms were measured in order T-3 3100 Hz, simulated T-3 520 Hz, and simulated T-3 3100 Hz; 1.10, 1.05, 0.73 and 0.65, respectively. In the case of older males, it was confirmed the high-beta increased with all fire alarms, similar to young females. The bell showed the largest increase at 1.98, and slow whoop showed the smallest increase at 0.26. For other fire alarms, T-3 3100 Hz, simulated T-3 520 Hz, T-3 520 Hz, and simulated T-3 3100 Hz measured at 1.83, 1.41, 1.05 and 0.87, respectively. For older females, T-3 3100 Hz increased at −0.62, but all other fire alarms showed decreased tendency. Specifically, simulated T-3 3100 Hz measured the largest decrease at −0.49, followed by bell, slow whoop, simulated T-3 520 Hz and T-3 520 Hz at −0.41, −0.26, −0.24 and −0.07, respectively. Statistical analysis of changes in high-beta showed no significant main effects in all variables. The interaction between age and gender (F: 30.520, *p* < 0.000) showed only significant results with large effect size (partial ŋ^2^: 0.094). As a result of post-hoc analysis, there were no significant differences in all variables.

#### 3.3.4. EEG Data: Gamma (30–50 Hz)

Finally, the gamma power according to the type of fire alarms, age and gender indicates that gamma power increased for young adults regardless of the type of fire alarm and gender. Specifically, young males showed the largest increase at T-3 520 Hz (0.96) and the smallest increase at T-3 3100 Hz (0.04). Other fire alarms measured, in order of bell, simulated T-3 520 Hz, simulated T-3 3100 Hz, and slow whoop, at 0.50, 0.39, 0.37 and 0.29, respectively. Young females also measured the largest increase at simulated T-3 3100 Hz (2.26) and the smallest increase at simulated T-3 520 Hz (0.89). Other fire alarms measured, in order of T-3 520 Hz, T-3 3100 Hz, slow whoop, and bell, at 1.87, 1.65, 1.24 and 0.98, respectively. Older males, similar to young adults, tended to have increased gamma power on all fire alarms. The highest increase was 3.19 at bell, and the smallest increase was 1.08 at slow whoop. The other fire alarms, which are T-3 3100 Hz, simulated T-3 520 Hz, T-3 520 Hz, and simulated T-3 3100 Hz, were measured at 2.80, 2.69, 1.95 and 1.18, respectively. In older females, slow whoop was decreased only at −0.24, but the other fire alarms were increased. Specifically, T-3 3100 Hz measured the highest increase at 1.86, followed by simulated T-3 3100 Hz, T-3 520 Hz, simulated T-3 520 Hz and bell at 0.77, 0.67, 0.61, and 0.11, respectively. The statistical analysis of changes in gamma power did not show main effects in all variables, similar to high-beta. In terms of interaction, a significant interaction between age and gender (F: 12.074, *p* < 0.001) was confirmed with *p*: 0.039). Additionally, post-hoc results were not found to be significant.

## 4. Discussion

The purpose of the present study was to investigate the subjective (psychoacoustic recognition) and objective (EEG) evaluations of listening to fire alarms, and also to identify the relationship between acoustic characteristics and measured responses. The subjective evaluation was conducted via a questionnaire on three factors: arousal, urgency, and immersion. The EEG was subdivided into theta, alpha, beta (SMR, mid-beta, and high-beta) and gamma power.

### 4.1. Psychoacoustic Recognition

In the present study, differences in arousal were identified depending on the acoustic characteristics of the fire alarms by measuring arousal, urgency and immersion. As a result of the main components of the developed questionnaire, question #2 was considered as arousal, but the factor was assigned to urgency. These results mixed the meaning of arousal and urgency. Other components such as urgency and immersion were assigned statistically as appropriate variables.

#### 4.1.1. Arousal

As a result of arousal evaluation, young males reported high arousal at a gradually increased frequency (slow whoop) or high frequency (bell, T-3 3100 Hz, simulated T-3 3100 Hz). The low frequency (T-3 520 Hz and simulated T-3 520 Hz) was evaluated as the low arousal. On the other hand, the young females rated T-3 520 Hz, which had low frequency, as high arousal, and likewise high frequency. In other words, young males tended to have higher arousal when the frequency of the fire alarm increased, but these tendencies were not confirmed by females. In the case of older adults, they also were evaluated to higher arousal when the frequency of the fire alarms increased, regardless of gender. These results were consistent with Gonzlez et al.’s (2012) results [49]. They reported that the higher frequency and higher intensity, the more arousal and annoyance with static correlation. In addition, induced arousal was reported to be immediate and consistently associated with urgency [19]. That is, the high-frequency fire alarms with high arousal suggest that a high urgency may also be observed. In contrast to the present results of high arousal at high frequency, a study by Buck et al. (2009) found that the T-3 520 Hz, which had a low frequency, was most effective for waking up sleeping residents [15]. Thus, it means that T-3 520 Hz suggests induced high arousal for sleeping residents. In terms of presbycusis simulation, the present study identified that T-3 (520 and 3100 Hz) showed significantly higher arousal than simulated T-3 (520 and 3100 Hz). Although these significant results encompassed only females, it suggests that the hearing thresholds of the listener affected induced arousal. These results are consistent with previous studies [10,15]. Additionally, the present study assumed that the older adults showed a disadvantage for hearing high-frequency fire alarms with low arousal. However, the older adults were highly aroused at high-frequency fire alarms, similar to young adults. In other words, the effects of presbycusis were not identified. Although the older adults have high-frequency hearing loss, also known as presbycusis, they were sufficiently aroused. However, these results are difficult to extend to the hearing loss group with moderate or greater hearing loss, especially at high frequency.

#### 4.1.2. Urgency

In the case of urgency, slow whoop was evaluated as high urgency regardless of gender. Similar to arousal, young males evaluated high urgency at gradually increased frequencies or high frequency (bell, simulated T-3 3100 Hz and T-3 3100 Hz). The low frequency (T-3 520 Hz and simulated T-3 520 Hz) evaluated low arousal. In contrast, although the T-3 3100 Hz was evaluated as high urgency, other high-frequency fire alarms (bell and simulated T-3 3100 Hz) were evaluated as relatively low urgency. Additionally, T-3 520 Hz, low frequency, showed relatively high arousal. In other words, the tendency that high frequency corresponded to high urgency was not found in young females. Older adults showed a tendency to have higher urgency when the fire alarms’ frequency gradually increased, regardless of gender. Additionally, older adults showed lower urgency than young adults for all fire alarms. In simulated T-3 (520, 3100 Hz), which simulated presbycusis, there were no significant differences for young males, but significant differences were confirmed for young females. This means that urgency can also be affected by the hearing thresholds. Consistent with previous studies, bell, slow whoop and T-3 3100 Hz with high frequency or gradually increased at low to high frequency were evaluated as high urgency [11,18]. In addition to those acoustic characteristics, the familiarity and identification of fire alarms can influence the urgency perception. The study by Lee [18] also reported high urgency with a siren that is similar to a slow whoop. However, the siren was evaluated as high urgency as a warning signal, not specifically a fire alarm. That was explained because it is recognized as a representative and familiar warning signal. In other words, not only increasing the frequency to increase the urgency, but also listening to the fire alarm should be able to induce an appropriate urgency and evacuation action in a fire situation. Substituting these results for the T-3, the T-3 may not be effective as a fire alarm in terms of familiarity and identification. Although the T-3 was recommended internationally, there were no standardized acoustic characteristics. In the studies by various researchers [9,16,17], it is also reported that the T-3 was recognized as a mechanical sound rather than a fire alarm. Therefore, awareness of the T-3 and the fire situation will be required through education and campaigns on the T-3 pattern, while the use of high frequency could induce a higher urgency.

#### 4.1.3. Immersion

Regarding immersion, immersion was relatively less affected by the type of fire alarm than arousal and urgency. Specifically, immersion commonly increased in slow whoop, but the difference according to the frequency was not noticeable when compared to arousal and urgency. In addition, immersion was the lowest in simulated T-3 520 Hz, but the similar fire alarm was not found to have a similar tendency in T-3 520 Hz. These results suggested that the acoustic characteristics between the two fire alarms were similar, but the perceived psychoacoustic difference was confirmed. Interestingly, young adults evaluated the highest immersion, while the elderly rated it with a relatively low immersion. In addition, young adults with simulated T-3 3100 Hz evaluated low immersion, while older adults evaluated it high immersion. In other words, the frequency of the fire alarm considered did not significantly affect immersion, and the difference according to age was confirmed. As a research design, the present study set immersion as a variable that can influence arousal and urgency. In other words, high immersion could enhance arousal and urgency [24,25]. Not only the frequency of the fire alarm, but also the high immersion was induced by slow whoop, which gradually increased frequency. Conversely, a monotonous fire alarm with low frequency, simulated T-3 520 Hz, induced low immersion. Considering that the definition of immersion was mental and physical excitement due to external stimuli [24], a monotonous pattern with a low-frequency fire alarm may not be sufficient to induce immersion. Moreover, listening to fire alarms in the present study was a passive listening condition, rather than interacting with the stimuli. In terms of presbycusis simulation, the young males showed significant differences between T-3 520 Hz and simulated T-3 520 Hz. The young females showed a significant difference between both 520 Hz and 3100 Hz. That is, as with arousal, this means that urgency was affected based on the listener’s hearing thresholds. In the case of older adults with normal hearing, they also tended to evaluate high urgency even in high-frequency fire alarms; the effects of presbycusis were not identified.

### 4.2. EEG Data

#### 4.2.1. Theta effect

As a result, for theta measurement, young males showed overall decreased theta power, but, in contrast, adult females showed increased theta power. In the case of older adults, they showed decreased tendency regardless of gender. In other words, except for young males, theta power was increased in listening to fire alarms. Moreover, theta power cannot identify the common feature according to the type of fire alarm, age, and gender. Statistically, no significant effect was found according to the type of fire alarm, age and gender; only interaction between gender and age was confirmed. Theta was often interpreted as associated with sleep, cognitive load, working memory and learning [31,32,33,34]. However, the purpose of the present study was to measure psychological changes by listening to fire alarms. Therefore, it may be desirable to focus on changes in emotion rather than on sleep, cognitive load, or working memory. In addition, the anterior cingulate cortex, which involves emotional processing, was located in the central frontal lobe [35]. The results of the theta distribution in this study also confirmed that the theta varied predominantly in the frontal lobe. In other words, listening to the fire alarm induced emotional changes. As pointed out earlier, overall theta power increased, except among young males, in listening conditions. In terms of emotional arousal, higher arousal was reported as probably necessary for the theta to occur [50]. Although the present theta results cannot identify the common features in theta changes, emotional changes such as arousal or urgency can be assigned to the increased theta power.

#### 4.2.2. Alpha Effect

In the present study, a decrease in alpha power was confirmed in all subjects regardless of age and gender. Considering the higher frequency of the fire alarm, the higher arousal and urgency, the younger group showed a relatively large decrease in alpha power at the high-frequency fire alarms (bell, T-3 3100 Hz and simulated T-3 3100 Hz). On the other hand, the alpha power of the slow whoop or T-3 520 (simulated T-3 520 Hz) was relatively greater in the older group. Statistically, the effect on the type of fire alarm was not significant, but it was confirmed to be a significant effect depending on age (young adults > older adults). In general, increased alpha power was often associated with comfortable conditions such as rest and relaxation. In other words, the decreased alpha was known to have a negative correlation with brain activity such as cognitive load, working memory [41]. Similar to theta, the association with emotions was also confirmed to decrease in negative emotions [42,43]. As a result of alpha power measurement in the present study, the decreased alpha power in all fire alarms was considered to be caused by negative emotions (e.g., nervousness, anxiety) following listening to fire alarms. Similarly, a study investigated the variation of alpha power by listening to pleasant (classic) and unpleasant (5000 Hz pure tones and teeth-gnashing) sounds. They reported that the alpha power was reduced in unpleasant sounds [51]. When interpreted with the psychoacoustic recognition, the tendency that the alpha pore decreased at high-frequency fire alarms corresponded to the psychoacoustic results; high arousal and urgency at high-frequency fire alarm. That is, it can be determined that alpha power decreased due to the higher arousal and urgency. However, these results were not consistent between subjective responses and EEG for the older group.

#### 4.2.3. Beta Effect

For SMR (Sensorimotor rhythm), decreased tendency in listening condition was confirmed, except for older males. In addition, the SMR also decreased more in high-frequency fire alarms, and decreased less in low frequency. In terms of beta power, the present study assumed that increased beta power induced negative emotions. However, the results of SMR power, which is a sub-category of beta power, showed the opposite results. In other words, SMR waves tended to decrease as the arousal and urgency increased. These results considered that the SMR power was reported to coexist with the somatosensory cortex, which was one of the resources of the alpha power [52]. Moreover, the frequency band of SMR power (12–16 Hz) was also partially shared with alpha power (8–13 Hz). In other words, decreased SMR power was considered due to these reasons. In addition, the traditional studies often interpreted increased beta power as a response to increased nervousness, anxiety and cognitive load [31,32,33,34]. However, the traditional interpretations focused on interaction with external stimuli with subjects such as 3D games or audiovisual media [44]. Considering that the present experiment passively presented only auditory stimuli, there was also a possibility that the cognitive load was not induced even though negative emotions were induced. 

For mid-beta, also, decreased pattern in listening conditions was confirmed in young males and older females. On the other hand, an increased pattern in listening conditions was confirmed in young females and older males regardless of the type of fire alarms. In other words, mid-beta was considered to be strongly affected by age and gender. However, the common features according to the type of fire alarm, age and gender were not found, with no significance. Previous studies that identified changes of mid-beta in response to external stimuli reported associations between tension arousal and mid-beta power. However, the present study cannot identify consistent mid-beta variability in response to fire alarms. The fire alarm aims to induce high arousal and urgency. By presenting unpleasant sounds, fire alarms induce the listener’s cognitive arousal and perceived risk. Therefore, fire alarms cause unpleasant emotions for the subjects. Similar to the mid-beta results in this study, Nishifuji and Miyahara [53] and Nishifuji et al. [51] studies reported that only alpha power was significantly decreased, and no significant change was founded in the beta power. On the other hand, other previous studies that reported an increase in beta power due to external stimuli [42,44] have shown interactions between external stimuli and subjects, such as audiovisual media, and games. Therefore, it is considered that the rather decreased mid-beta induced negative emotions in the subjects but did not require interaction with the listener nor demanded cognitive resources. 

Similar to the mid-beta, high-beta also tended to be decreased in young males and older females in listening conditions, while increased in young females and older males. Among them, the high-frequency fire alarm that induced high arousal and urgency decreased the most for young males and older females and showed the highest increase for young females and older males. However, the low-frequency fire alarm did not show consistent results with psychoacoustic recognition, such as a large decrease or a small increase at T-3 520 Hz. One interesting result is that as the frequency band of EEG increases from SMR to high-beta, the overall decrease tendency which showed in the early beta; SMR turned into an overall increased tendency at high-beta. Even though the present study did not confirm a consistent increasing tendency in beta power through listening to fire alarms, it is expected that the more interaction between subjects and external stimuli or demands for a higher cognitive load, the more consistent the results will be with the hypothesis.

#### 4.2.4. Gamma Effect

As regards gamma power, it generally tended to increase in listening conditions. However, no common features or significant differences were found. Generally, the gamma power often reported being induced during extreme arousal, anxiety, excitement, tension, and high cognitive performances. Therefore, it often occurs dominantly in the frontal lobe in arousal conditions, and the increased tendency was reported when excited or nervous [44]. Although no significant difference was found between fire alarms in this study, the overall increased gamma power in listening conditions was considered to be affected by arousal and urgency. Similar to the present study, Nishifuji et al. investigated the changes in gamma power by listening to unpleasant sounds and did not find a significant difference in gamma power, only a significant difference in alpha power [51]. That is, they argued that unpleasant emotions were only related to alpha power. Although these results were limited in comparing directly with the present study due to methodological differences, the results of Nishifuji et al. were consistent with the present study [51].

### 4.3. Limitations of the Present Study

Several limitations in the present study warrant further research. First, the present study did not explain the correlation between EEG and each factor of psychoacoustic recognition. In addition, since the changes of EEG apply to various factors and complexes, they do not directly correspond with the listener’s psychological experience including various factors affecting the cognitive response to the six-alarm stimuli. For example, sound could be recognized by participants as a slightly different alarm based on their previous experience. Thus, we suggest subdividing and verifying the arousal, urgency, and immersion with the correlation between the subdivided factors and EEG in the future study while explaining the simple–complex network of the cognitive responses.

Second, there was the structural limit of the laboratory setting in the method design. Presenting fire alarms to subjects who know that it is a laboratory setting might affect the response of the subjects. For example, for the subjects who are aware that they are participating in an experiment, listening to fire alarms can be substantially non-threatening. Therefore, further studies require methodological complementation via a more realistic environment (e.g., accompanying visual stimulus) and extend to detailed design while having different background noise levels, considering dimensions of the alarm sounds, and comparing factors by indoor and outdoor environments.

Third, the simulated T-3 (520 and 3100 Hz) fire alarms, which simulated presbycusis, showed significantly lower arousal and urgency to the younger groups. However, the older group with normal hearing showed enough high arousal and urgency at high-frequency fire alarm. This means that older adults with normal hearing were able to respond to high-frequency fire alarms. Therefore, it is difficult to extend the results of older adults to the hearing-impaired group with severe hearing loss or more than moderate hearing loss at high frequency. In terms of the intensity of fire alarms, the psychoacoustic response can also be affected by the intensity. In the present study, we only investigated the effects of frequency; the intensity of the fire alarm was not investigated. However, in an actual fire situation, effects such as high background noise, distance between fire detector to bedroom, and transmission attenuation process were not considered; we used fixed 75 dBA based on guidelines [7,8]. Additionally, the correlation between psychoacoustic recognition and EEG cannot found significant effects. For the following research, not only the frequency of the fire alarm, but also intensity including specific stimulus of each individual such as medium tonal hearing loss should be considered in terms of signal-to-noise ratio, and transmission attenuation.

Even given these several limitations, the present study newly tried to analyze subjective responses according to the acoustic characteristics of a total of six fire alarms. In addition, by measuring EEG, the psychological changes were studied objectively and contributed to the methodological elaboration.

## 5. Conclusions

The present study investigated the relationship between the acoustic characteristics of the fire alarms and subjective response (psychoacoustic recognition) and objective measurements (EEG) in young and older adults with normal hearing. When considering its purpose, the fire alarms with high-frequency or gradually increased frequency such as T-3 3100 Hz, bell, slow whoop can be effective because they induced high arousal and urgency. Regardless, the high frequency causes high arousal and urgency, but cannot be widely transmitted and are also vulnerable to background noise. Thus, it would be better to use fire alarms by separating their acoustic characteristics according to the place of use and detailed purposes although it should be supported by scientific evidence of the following study. For example, high-frequency fire alarms are suggested for use in a private home or bedroom. Conversely, since the low-frequency fire alarms have a wide transmission range and are advantageous for diffraction and masking, they are appropriate for use in an outdoor fire hydrant in apartments or public places.

In addition, the internationally recommended T-3 pattern has various responses depending on the frequency of the fire alarm. However, the advantages of the T-3 pattern confirmed no significant effect. In a similar temporal pattern, slow whoop with gradually increasing frequency band was evaluated as high arousal and urgency in spite of the frequency band of about 500 to 1500 Hz. In other words, it is expected there will be advantages of low frequency with high arousal and urgency. Unfortunately, no significant difference was found in all EEG depending on the type of fire alarms. Nevertheless, when combined with the analysis of the psychoacoustic recognition, the increased frequency of the fire alarm corresponded to more negative emotions with the decreased tendency of alpha power. However, theta was not consistent; rather, the increased tendency was confirmed. In the case of beta power, it was expected that would be increased due to arousal, tension or nervousness, but instead showed a tendency to decrease overall. Finally, gamma power was associated with tension, excitement, and anxiety, and it tended to increase overall.

## Figures and Tables

**Figure 1 ijerph-18-00541-f001:**
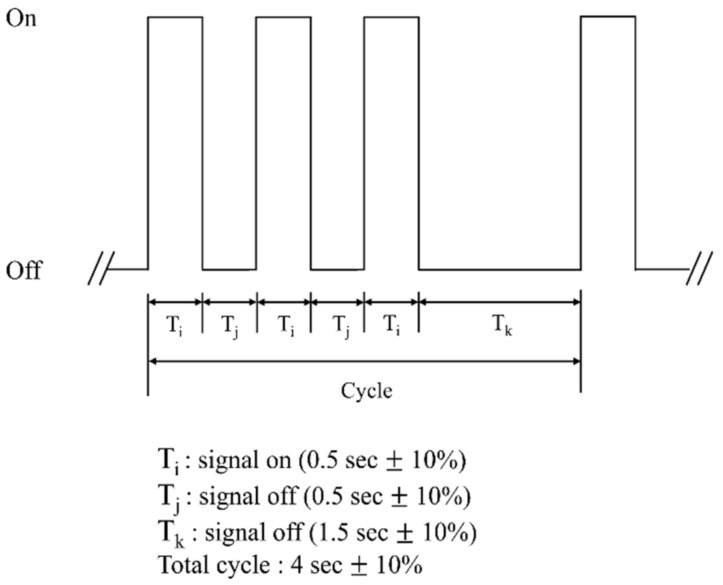
An example of temporal three (T-3) pattern.

**Figure 2 ijerph-18-00541-f002:**
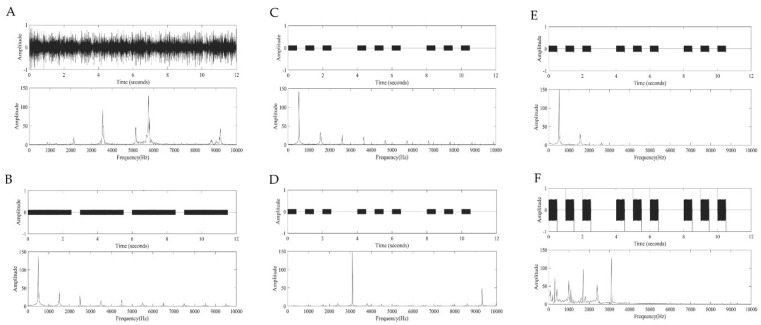
A screenshot of spectrum analysis for the six fire alarm sounds applied in the present study: bell (**A**), slow whoop (**B**), T-3 520 Hz (**C**), T-3 3100 Hz (**D**), simulated T-3 520 Hz (**E**), simulated T-3 3100 Hz (**F**). Each panel had an amplitude of two aspects for the time domain (upper) and frequency domain (bottom).

**Figure 3 ijerph-18-00541-f003:**
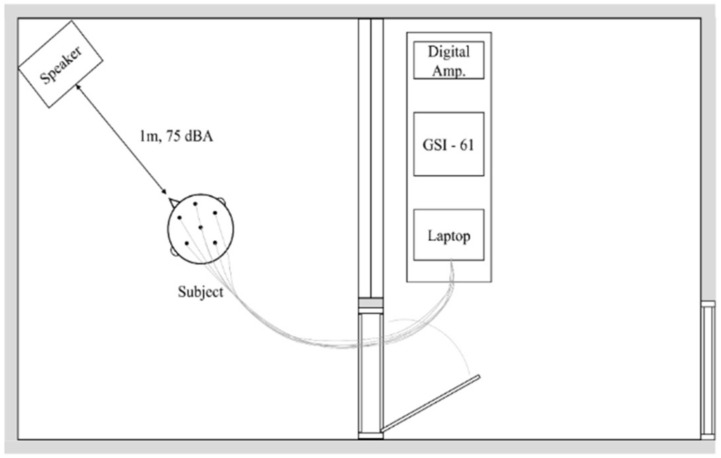
An image of testing conditions when recording electroencephalography.

**Figure 4 ijerph-18-00541-f004:**
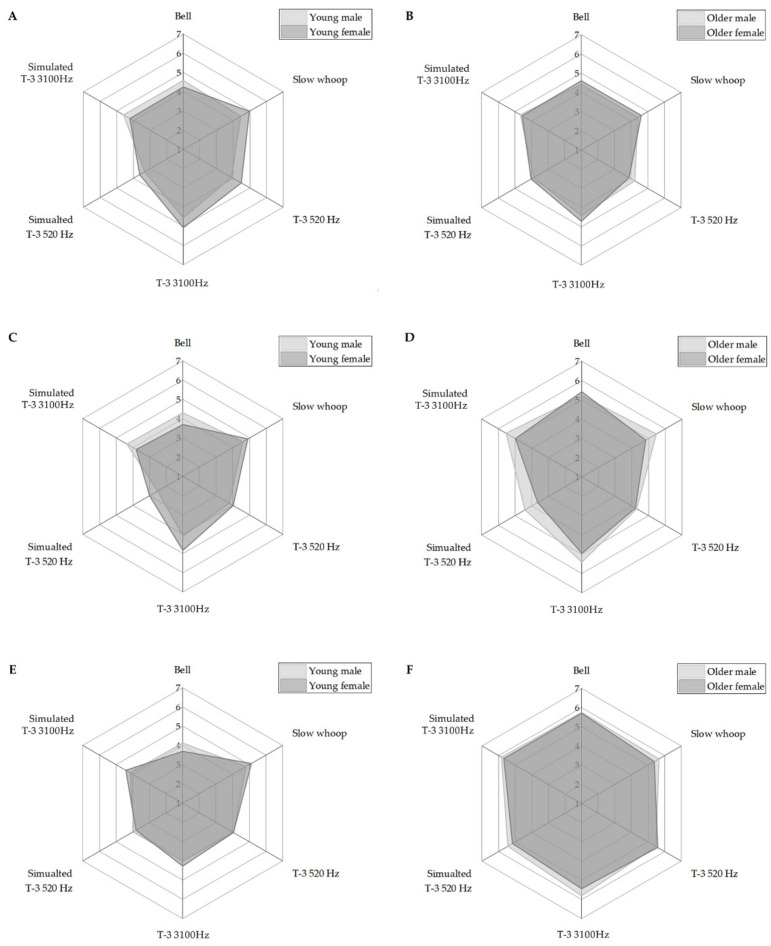
Results of psychoacoustic recognition induced by six different types of fire alarm sounds including age factor (left panels for young and right panels for old) and gender factors (light gray for male and dark gray for female): arousal (**A**,**B**), urgency (**C**,**D**), and immersion (**E**,**F**).

**Figure 5 ijerph-18-00541-f005:**
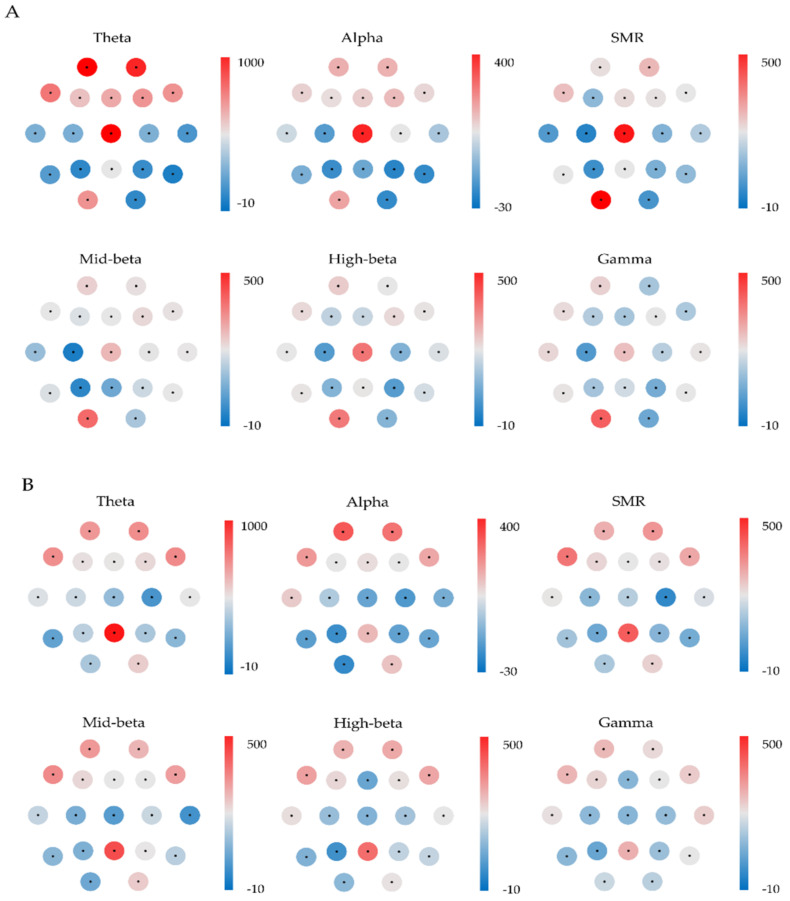
Electrode usage of six electroencephalography (EEG) subgroups is indicated by the red and blue of the circle around each electrode location at (**A**): T-3 3100 Hz, (**B**): T-3 520 Hz that showed significant differences in psychoacoustic recognition.

**Table 1 ijerph-18-00541-t001:** Hearing information between young and older adults as group mean and standard deviation.

	Age (Year)	Hearing Thresholds in Frequency (Hz)							MCL	UCL
		250	500	1000	2000	4000	8000	PTA		
YG	25.1 (1.58)	3.80 (5.05)	4.00 (5.95)	2.80 (4.35)	4.40 (4.40)	5.60 (4.40)	5.40 (4.54)	4.20 (4.86)	58.39 (3.86)	96.60 (5.28)
OG	74.48 (3.70)	20.20 (7.43)	20.80 (9.32)	23.40 (8.63)	28.00 (11.09)	38.60 (17.05)	51.80 (12.41)	27.70 (13.64)	67.58 (4.97)	105.32 (4.46)

YG: Young adults group, OG: Older adults group, PTA: Pure-tone average of 500, 1000, 2000, and 4000 Hz, MCL: Most comfortable level, UCL: Uncomfortable level.

**Table 2 ijerph-18-00541-t002:** Six different types of fire alarm sounds used in the present study and their acoustic characteristics.

Stimulus	Fundamental Frequency	Time Interval
Bell	Main frequency: 3533, 5149, 5752 Hz	Steady
Slow whoop	500 Hz base frequency and gradually rises approximately one octave	3.5 s ON/500 msec OFF
T-3 520 Hz	512 Hz with the odd harmonics (3rd, 5th, 7th, etc.)	500 msec ON/500 msec OFF
T-3 3100 Hz	3100 Hz with the odd harmonics (3rd, 5th, 7th, etc.)	500 msec ON/500 msec OFF
Simulated T-3 520 Hz	T-3 520 Hz with deteriorated high-frequency	500 msec ON/500 msec OFF
Simulated T-3 3100 Hz	T-3 3100 Hz with deteriorated high-frequency	500 msec ON/500 msec OFF

**Table 3 ijerph-18-00541-t003:** The results of three-way analysis of variance (ANOVA) among signal, age and gender in arousal.

Independent Variable	Variable	Type III Sum of Squares	df	Mean Square	F	Partial ŋ^2^
Arousal	Signal	517.940	5	103.588	6.845 ***	0.110
	Age	0.960	1	0.960	0.063	0.000
	Gender	25.667	1	25.667	1.696	0.006
	Signal × Age	74.864	5	14.973	0.989	0.018
	Signal × Gender	57.647	5	11.529	0.762	0.014
	Age × Gender	8.427	1	8.427	0.557	0.002
	Signal × Age × Gender	58.197	5	11.639	0.769	0.014
	Error	4177.103	276	15.134		
	Total	0.000	300			

Note: *, *p* < 0.05; **, *p* < 0.01; ***, *p* < 0.001.

**Table 4 ijerph-18-00541-t004:** The results of three-way ANOVA among the type of fire alarm, age and gender for urgency.

Independent Variable	Variable	Type III Sum of Squares	df	Mean Square	F	Partial ŋ^2^
Urgency	Signal	1315.687	5	263.137	10.119 ***	0.155
	Age	958.043	1	958.043	36.843 ***	0.118
	Gender	26.995	1	26.995	1.038	0.004
	Signal × Age	213.084	5	42.617	1.639	0.029
	Signal × Gender	59.767	5	11.953	0.460	0.008
	Age × Gender	62.043	1	62.043	2.386	0.009
	Signal × Age × Gender	199.404	5	39.881	1.534	0.027
	Error	7176.910	276	26.003		
	Total	103,730.000	300			

Note: *, *p* < 0.05; **, *p* < 0.01; ***, *p* < 0.001.

**Table 5 ijerph-18-00541-t005:** The results of three-way ANOVA among type of fire alarm, age and gender for immersion.

Independent Variable	Variable	Type III Sum of Squares	df	Mean Square	F	Partial ŋ^2^
Immersion	Signal	191.550	5	38.310	2.867 *	0.049
	Age	2438.144	1	2438.144	182.481 ***	0.398
	Gender	7.847	1	7.847	0.587	0.002
	Signal × Age	196.966	5	39.393	2.948 *	0.051
	Signal × Gender	26.270	5	5.254	0.393	0.007
	Age × Gender	8.917	1	8.917	0.667	0.002
	Signal × Age × Gender	72.592	5	14.518	1.087	0.019
	Error	3687.654	276	13.361		
	Total	0.000	300			

Note: *, *p* < 0.05; **, *p* < 0.01; ***, *p* < 0.001.

## Data Availability

Not applicable.

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
