# Peer review of "Changes in Psychoacoustic Recognition and Brain Activity by Types of Fire Alarm"

_ijerph, 2021, doi:10.3390/ijerph18020541_

Round 1
Reviewer 1 Report
The methodology is correct, and the instruments are adequate. The statistical analysis was also rigorous and adequate.
However, several points as indicated below need to be addressed by authors to improve the quality of the article:
- Which tests have been done on the patients? Skills like cognition, memory-spam etc was tested in the patients? I think it is important more cognitive details on this evalution.
- In the introduction you mention " the high-frequency fire alarm is not appropriate for listeners with hearing loss, especially poor hearing ability in the high frequencies due to aging." (72-74). How is it defined and how was the hearing of the patients included in this study? You mention “normal hearing thresholds in the testing frequencies of 250 to 8,000 Hz as a function of age” (152-153), but it will be interesting to characterize the hearing thresholds for both groups in the sample.
- In the inclusion criteria (151-153), shouldn't they have considered absence of neurological, psychiatric disease and taking medication? These variables can influence the EEG responses, so it is important to include these criteria.
- As mentioned in the lines 175-177,” the older adults have higher thresholds in high frequencies than their young hearing counterparts even though they have normal hearing for their age”. The intensity of the stimulus took into account the hearing thresholds according the age? As mentioned the intensity level of the fire alarm was 75 dBA (224-225). This intensity may be high for the younger group, the same may not happen for the older group of people due to the effect of aging on hearing - presbycusis. Late you mentioned “This means that urgency can also be affected by the hearing thresholds” (576-577). Why not consider a specific stimulus for each individual, for example, 50dB above the Medium Tonal Hearing Loss?
- The results are detailed, but do not explain the correlation between the EEG and each psychoacoustic recognition factor: excitement, urgency and immersion, as assumed in the limits of the study.
- In the line 549 you mentioned “higher the frequency (or intensity)”, you mean said higher the frequency and higher intensity?
- You mentioned “for future research, not only frequency of the fire alarm, but also intensity should be considered in terms of signal-to-noise ratio, and transmission attenuation.” (732-733), I propose to consider also the Medium Tonal Hearing Loss.
Author Response
Thank you very much for your valuable and significant comments. Based on the comments received, we had discussed several times and understood the reviewer’ multiple concerns. Our paper changed into a better version while considering all comments which you pointed out. Please see our response in the attached file and find newly changed parts of red letters in our revised manuscript. Again thanks.

Reviewer 2 Report
The aim of this study was to associate the EEG activity and the Psychoacoustic Recognition induced by an alarm in young and old subjects.
While potentially, the article may have interest to the IJERPH readers, the article contains a lot of data (to my view too much), which makes it very difficult to keep track of the main findings.
The issue is very complicated and contains a lot of factors that can affect the cognitive response to an alarm. Of them, the age of the subjects, cognitive health, presence of hyperacusis especially in the old subjects, six different types of the alarms, all can affect the response of the individual to an occasional alarm.
Thus, my immediate suggestion is to reduce significantly the length of the article to the most important factors and to focus on the most important findings.
Other comments:
- How the authors made sure to calibrate the loudness across all different types of the stimuli?
- The FFT of the presented signals must be presented.
- Old subject may suffer from loudness recruitment. Did they check for that??
- How was the presentation order of stimuli, random, sequential???
- What kind of task the subject had during the presentation of the stimuli?? Since h
- What actions/instructions were taken to avoid head movements, since they affect intensity and spectrum reaching the ears.
- The figures must be simplified
Author Response

(The authors gave the same response as above.)

Reviewer 3 Report
This paper investigates the relationship between the acoustic characteristics of fire alarm sounds and human psychological and physiological responses to them. The results are a little unclear, but it is concluded that alarm sounds with high frequency components are more effective in giving people a sense of urgency.
There are enough citations to previous studies, and the analysis method is modern. There are no major problems. However, I think that additional explanations are needed for some points. Answering the following comments is requirement for publication.
(1) Line 159: 2.2. Development of 6 types of fire alarm stimuli
State whether or not each stimulus is a sound that could be recognized by participants as an alarm sound based on their previous experience. As the authors point out, sounds that are associated with certain information may produce a conditioned reflex response, independent of their acoustic properties.
(2) Line 236: … with a 3-minute break between …
State whether the measurement of the reference EEG (background EEG) was taken before each alarm sound or only once before listening to the six alarms. If the latter, explain whether the 3-minute break stabilized the EEG data to the point where it could be considered similar to the reference EEG.
(3) Line 745: Thus, the high-frequency fire alarms are …
There may be a problem with this suggestion. I believe that, ideally, the same alarm sound should always be used for a given event in order to encourage quick evacuation action. It seems that additional explanation is needed, such as changing only the frequency characteristics within the range that is recognized as the same alarm sound.
Author Response

(The authors gave the same response as above.)
